# Spontaneous Officinal Plants in the Cilento, Vallo di Diano and Alburni National Park: Tradition, Protection, Enhancement, and Recovery

**DOI:** 10.3390/plants12030465

**Published:** 2023-01-19

**Authors:** Enrica De Falco, Daniela Rigano, Vito Fico, Antonella Vitti, Gaia Barile, Maria Pergola

**Affiliations:** 1Degree Course of Agriculture, Dipartimento di Farmacia, Università degli Studi di Salerno, Via Giovanni Paolo II, 132, 84084 Fisciano, Italy; 2Dipartimento di Farmacia, Università di Napoli Federico II, Via Domenico Montesano 49, 80131 Napoli, Italy; 3Associazione “Sanza Città della Lavanda”, 84030 Sanza, Italy

**Keywords:** medicinal plants, rural development, sustainability, ethnobotany, traditional drying

## Abstract

The aim of this study was to deepen our knowledge on the heritage and traditional uses of some medicinal plants of the Cilento, Vallo di Diano and Alburni National Park (Salerno province) and to evaluate their productive potential, in order to increase possible uses to recover and enhance the territory. Biometric surveys and biomass evaluation were carried out. Two types of aqueous extract were prepared using air-dried samples of six harvested species and tested for anti-germination activity on *Lepidium sativum* L. Hydrolates were recovered via steam distillation from aromatic species and the chemical–physical characteristics were determined. Historical evidence of industrial activity was collected in the territory of Sanza on Monte Cervati, where lavender essential oil has been distilled in the past century, and characterization of the essential oil components was carried out. The ethnobotanical uses detected mainly concerned traditional medicine and nutritional, ritual, or religious uses. The experimental results highlight that spontaneous medicinal plants could become potential sources of local economic development, with uses not only in the phytotherapeutic sector, but also in others, such as food and agriculture for weed control. Moreover, the evidence derived from industrial archeology could represent a further driving force for the enhancement of the territory’s resources.

## 1. Introduction

The importance of biodiversity began to be recognized worldwide due to the Rio Convention in 1992 [1]. Since then, the European Union (EU) has enacted various regulations in order to safeguard and protect biodiversity. Among these, the most recent are Regulation n° 1143/2014 [2], which lays down provisions that aim to prevent and manage the introduction and spread of invasive alien species; Commission Delegated Regulation n° 531/2015 [3] on the protection and restoration of biodiversity and marine ecosystems to mitigate climate change and increase the energy efficiency of fishing vessels; and Regulation n° 848/2018 [4] on the organic production and labeling of organic products.

The COVID-19 pandemic highlighted the link between our health and the natural environment and the need to protect biodiversity. Recovering the balance of ecosystems and safeguarding biodiversity are essential objectives for the prevention of future pandemics and the reduction of the impacts of climate change. In this context, the EU has published a new biodiversity strategy, which sets important and ambitious goals to be achieved by 2030 [5]. More than half of the global gross domestic product (GDP) depends on nature and the ecosystem services that it offers. On the other hand, plants have traditionally represented one of the major sources of resources to satisfy man’s multiple needs such as those of food, medicine, tools, clothing, buildings, and manufactured goods. In small rural settlements, the heritage of this knowledge, linked with the perception that the inhabitants of a community have of the environment in which they live, has been transmitted from generation to generation, mostly through oral testimonies. It is a vast but very fragile heritage, continuously threatened by the rapid socio-economic changes that accompany the disappearance of rural societies. Ethno-botanical research is extremely important because it helps to preserve and make known the uses for plants in different territories [6].

Traditionally, the harvesting of plants grown spontaneously in the natural environment represents a peculiar characteristic of medicinal and aromatic plants. However, collection activity, if not adequately regulated, can have environmental impacts such as biodiversity loss, the extinction of local species, and habitat degradation. In fact, it is estimated that 4000–10,000 different species of medicinal plant are currently at risk of extinction due to excessive harvesting [7]. At the same time, in Italy, the cultivation of officinal plants covers only 30% of the national needs [8]. The identification of spontaneous species of medicinal and aromatic interest represents an important option to protect natural biodiversity and the opportunity to introduce crops for products at sustainable prices.

The tradition of harvesting and using wild plants is still strongly rooted in many rural communities of the Cilento area (Campania region), as are the popular uses, traditions, and customs connected to them. A very vast territory is included in the Cilento, Vallo di Diano and Alburni National Park, declared a UNESCO World Heritage Site in 1998; this park is very rich in biodiversity and, in fact, contains many Sites of Community Importance (SICs) [9] belonging to the Natura 2000 network. For these reasons, over the years, it has been involved in research and projects with the aim of deepening our knowledge of the traditional use of plants. Scherrer et al. [10] documented the local use of 90 different plant species for medicinal, food, and domestic purposes in two distinct regions of the Cilento National Park (Monte Vesole and Ascea). Salerno and Guarrera [11] conducted research on ethnobotanical uses in Castel San Lorenzo. Mancini et al. [12] conducted a study on ethnobotanical uses in Vallo di Diano, which led to the identification of 152 plants belonging to 59 families, used for different purposes. In addition, De Falco and Di Novella [13] investigated the potential of spontaneous plants in the park area as dyeing plants to meet the demand for products and production processes with low environmental impact. Referring to spontaneous alimentary plants, De Falco et al. [14] reported the potential of spontaneous edible plants collected in different areas of Cilento, among others in the Campania region, for use as functional foods. Moreover, the phytomorphological and essential oil characterization, “in situ” and “ex situ”, of wild biotypes of oregano, also collected in the Cilento, Vallo di Diano and Alburni Park, was reported by De Falco et al. [15]. Actually, aromatic plants also have an important contribution to quality cuisine, and this has increased the search for new species and varieties for cultivation and for techniques to improve the sustainability of processes. The studies conducted on the recovery of waste from the cultivation and processing of aromatic herbs highlighted the opportunity to recover essential oils and aromatic waters from the perspective of circular economy and “zero residues” [16].

Based on what has been reported above, the objective of this study is to deepen our knowledge on some spontaneous medicinal plants in an area of the National Park of Cilento, Vallo di Diano and Alburni (Monte Bulgheria) that have, until now, not been taken into consideration (as far as we know), in order to contribute to the enhancement and protection of the territory under study through the development of potential forms of local economy. For this reason, this research takes into consideration the traditional harvesting and uses of eight plants that are widespread in the area, to contribute, above all, to the preservation of this heritage of knowledge. Moreover, the current scientific interest in these plants is reported to support their potential cultivation. Finally, this research takes into consideration the determination of some physical–chemical parameters of aromatic water, residues of distillation, and aqueous extracts, and is useful in evaluating possible non-traditional and sustainable uses for the plants under study.

## 2. Materials and Methods

### 2.1. Collection of Samples, Identification, and Biomass Evaluation

Samples of eight species were harvested at Monte Bulgheria, chosen for its particular landscape and floristic values (Figure 1), as it is included in the SIC “Monte Bulgheria—IT8050023”, in the year 2020. 

The following species were collected: *Foeniculum vulgare* Mill., *Lavandula angustifolia* Mill., *Malva sylvestris* L., *Matricaria chamomilla* L., *Myrtus communis* L., *Origanum vulgare* subsp. hirtum Link (*O. heracleoticum* L.), and *Rosmarinus officinalis* L. (Figure 2). The species were chosen according to their consolidated use and diffusion in the study area; the choice was also made based on the possibility of using them in preparations accessible to small farms, such as aromatic waters for food use, food herbal teas, and antimicrobial perfumes, with a view to enhancing local resources.

The identification of the species was made based on their botanical characteristics, as reported by Pignatti [17]. The specimens for each species were kept at the Pharmacy Department of the University of Salerno (De Falco, Bulgheria 1–8). 

The harvesting took place in typical locations for each species, which were appropriately geolocated. For each species, the phenological stage was noted, the height of the vegetation and the ground cover were measured, and the number of stems per plant (limited to herbaceous plants) were marked. The percentage moisture content of the biomass at harvesting was calculated as (g fresh weight—g dry weight)/g fresh weight %, after drying the material in an oven at 70 °C until constant weight. All the measurements were taken with three repetitions. For each sample harvested, the total weight and the weight of the leaves and of the inflorescences were measured, and the relative percentages compared to the total were calculated.

### 2.2. Ethnobotanical Uses

The ethnobotanical uses of the species were investigated through interviews conducted with the inhabitants of Celle di Bulgheria, a small village immediately at the foot of Monte Bulgheria. Open interviews (20) were carried out, excluding information suspected of “cultural pollution”, i.e., coming from texts, magazines, television, and other mass media. Respondents were asked to indicate the uses of the plants under study in traditional medicine, in food, as flavorings, in crafts, in domestic use, and for ritual or religious purposes. In addition, for each species, the local name was recorded.

Furthermore, given the historical importance (until about 1960) of the industrial distillation of the lavender harvested on Monte Cervati (Sanza, SA, Italy), located a short distance from Monte Bulgheria, the available documentation on this activity was acquired through on-site inspections, photographic acquisitions, and interviews made available by the Association “Sanza Città della Lavanda”.

### 2.3. Recovery of Aromatic Waters

The aromatic species (*Foeniculum vulgare*, *Lavandula angustifolia*, *Matricaria chamomilla*, *Myrtus communis*, *Origanum heracleoticum*, and *Rosmarinus officinalis*) were submitted to steam distillation from fresh biomass to recover the aromatic waters (hydrolates), made up of the essential oil remaining in the distillation water. The aromatic waters, according to the literature, are composed of highly diluted acids, with a delicate and pleasant scent which recalls that of the plants of origin [18].

The biomass of the aromatic plants was cut into small pieces, and then, submitted to distillation for 3 h using a Clevenger extractor (50 g of plant and 100 mL of water) according to European Pharmacopoeia [19]. Each extraction was carried out in triplicate. The hydrolates were collected without preliminary separation of the essential oils. Moreover, residual waters from the steam distillation were recovered, as reported by Zaccardelli et al. [16], to evaluate the possibility of recovering process residues from circular economy perspective.

The aromatic water and the residual waters were stored at 4 °C in the dark and in sterile containers to avoid contamination and the development of microbial agents [20].

### 2.4. Conservation using Traditional Methods

For six species (*Hypericum perforatum*, *Lavandula angustifolia*, *Malva sylvestris*, *Myrtus communis*, *Origanum heracleoticum*, and *Rosmarinus officinalis*), the biomass, once harvested, was immediately dried to avoid the proliferation of microorganisms and the development of enzymatic reactions. The drying method used was the traditional one at air temperature, away from sunlight. Drying was carried out by placing the plants on a trellis to let the air pass freely; furthermore, the plants were turned over periodically until they were sufficiently dried.

After drying, the samples were placed in glass jars that had been previously sterilized via boiling and were stored in a cool place away from light to avoid possible oxidation reactions.

For each plant, the drying time was registered and the residual humidity of the samples recorded with three replications. Moreover, the presence of foreign materials, both vegetable and not, as well as molds or damage from parasites, was detected. To achieve this, three samples representative of the whole biomass were selected, each with a quantity of 30 g. 

For each sample, the color was described both via visual evaluation and according to the Munsell Color System by exposing the samples to a white lamp (Philips TLD 18 watt/96) in order to identify the hue, which indicates the contribution of the main shades (red, yellow, green, blue, and purple), the value (a measure of darkness or lightness of a color on a scale of 0–10) and chroma, the latter of which is the degree saturation of a color [21].

### 2.5. Extraction from Dried Material

For the six dried species, as previously reported, extraction was carried out using water as a solvent, to perform easily reproducible eco-friendly processes. Two types of extraction were performed: hot extraction and at air temperature. In the first case, the conditions were 100 °C and 2 h of extraction. The weight/volume ratio used was 1:5 between the plant and water, respectively. Thus, for each sample, 500 mL of hot water was added to 10 g of material. In the second case, the samples were left to extract at air temperature for 72 h away from light. Additionally, in this case, the weight/volume ratio used was 1:5. At the end of the extraction process, each extract was filtered, collected in dark glass bottles, and stored in the refrigerator. For each extraction, three repetitions were made.

### 2.6. Chemical–Physical Analysis

To detect other potential uses of the studied species, for both hydrolates (residual water and aqueous extracts), the pH and electrical conductivity (EC), expressed in μS/cm or mS/cm, were determined using a pH meter/PC tester conductivity meter (XS Instruments, Capri, MO, Italy).

Spectrophotometric analysis was performed using a Thermo Scientific Multiskan Spectrum dual beam spectrophotometer with version 2.2 software. The reading of the extracts was performed in 1 mL semi-micro PMMA UV cuvettes with a wavelength range of 250–700 nm.

### 2.7. Evaluation of the Anti-Germinative Activity

To verify the potential use of the studied species in other sectors related to agriculture and their ease of application at farm level, the anti-germinating activity test was carried out. It involved both aqueous extracts (hot and air temperature) using cress seeds (*Lepidium sativum* L.) characterized by rapid germination. For each test, 3 control Petri dishes (with water only) and 3 plates for each type of extract obtained were prepared. Five Whatman filter paper discs were placed on each plate. Subsequently, 7 mL of water was added to the control plates, while 7 mL of the liquid extract to be tested was added to the sample plates. To all plates, 10 seeds were added.

The plates were left to incubate in the dark at a temperature of about 24 °C for 72 h. The number of normally germinated seeds was counted, and the primary radicle lengths (root + hypocotyl) were measured using a precision digital caliper (Mitutoyo Digimatic caliper).

### 2.8. Characterization of the Essential Oil of Spontaneous Lavender

Given the interest in the economic value of lavender essential oil (EO), in accordance with the research objective to contribute to the valorization of resources for the development of the territory, we analyzed the composition of the EO of spontaneous lavender (*Lavandula angustifolia* Miller) collected around Monte Cervati (Sanza, SA, Italy) where, in the past, lavender was distilled for industrial purposes (Figure 3). The phytochemical composition of spontaneous lavender oil was compared to that of a commercial sample (Lavandeto of Assisi, Assisi, Italy).

Samples of lavender inflorescence were subjected to EO extraction, as reported in Rigano et al. [22]. Essential oils were separated using n-hexane as a solvent (Sigma-Aldrich, Milan, Italy) and yield percentage was calculated on the fresh total biomass. Then, they were kept at 4 °C in the dark. 

Essential oils analysis was performed via GC and GC-MS, as described in Formisano et al. [23]. The constituents were identified via GC and GC-MS analysis to compare their retention indices (Kis) with those present in the literature, or with those of standards available in our laboratories, as described before [22,23].

### 2.9. Statistical Analysis

All data are indicated as mean ± standard error. The data of the biological assay are expressed as a percentage relative to the control (%). The significant differences (*p* < 0.05) within the anti-germination activity were evaluated by means of one-way analysis of variance (ANOVA). Comparisons among the means were performed using the Tukey post hoc test (*p* = 0.01). The statistical analysis was performed using the MSTAT-C software package (Michigan State University, East Lansing, MI, USA).

## 3. Results

### 3.1. Collection of Plants and Biomass Relieves

The selected species are all widespread throughout the entire National Park, particularly in the reference area [24].

Table 1 shows the different locations of Monte Bulgheria where the collection was carried out for each species under study. 

The harvest was carried out with respect paid to balsamic time, the customs of the place, and any new uses. 

As can be seen in Table 1, the species were harvested between the third ten days of April and the last ten days of August. They were collected at the beginning of flowering or at full bloom, except for the fennel, which was harvested at the end of flowering—beginning of waxy ripening, and rosemary, which was harvested in the vegetative stage. In general, the weight of the leaves and stems had higher percentages than the flowers or inflorescences when present, except for fennel.

The moisture values at harvesting ranged from the minimum value of 59%, for myrtle, to the maximum value of 80.4% for mallow. 

As can be seen from the data in Table 1, the different development of plant cover is evidenced, in relation to the habitus of the species. For some of them, such as rosemary, lavender, and myrtle, the degree of cover (2776 m^2^, 1180 m^2^, and 1084 m^2^ per plant, respectively) was much higher than that of the other species taken into consideration. Similarly, the volume estimate per plant, calculated according to Hidalgo and Harkess [25], was found to be useful in highlighting differences in growth between species. The highest values were recorded for rosemary (3830 m^3^), followed by myrtle (1820 m^3^) and lavender (0.786 m^3^). These data can be useful to complete the information available on the different species concerning the mechanical characteristics of the roots [26,27].

### 3.2. Results of Ethnobotanical Research

Table 2 reports the synthesis for each species under study with their scientific names, their local names as revealed through the interviews, and their English common names. 

In the following subparagraphs are reported the botanical characteristics of the species under study and the most common uses according to the ethnobotanical research in the reference area.

#### 3.2.1. *Foeniculum vulgare* Mill.

The genus *Foeniculum* is native to the Mediterranean area. *F. vulgare* and was used as an aromatic and medicinal plant by both the Greeks and the Romans. All parts of the plant are aromatic. It is widespread in Central and Southern Italy up to altitudes of 1000 m a.s.l. It shuns alkaline soils, prefers loose and fertile soils, and does not tolerate water stagnation [28]. It is widely spread in the Park of Cilento, Vallo di Diano and Alburni [24] and it is reported among the dyeing plants of Cilento, Vallo di Diano and Alburni Park [13]. 

The dialect name reported from the interviews is *finùcchiu sirvàticu* (Table 2). From the interviews, it emerged that the fruits are still used to flavor soup, sausages, dried figs, and biscuits. The decoction of dried fruit is used as a digestive aid and a carminative. It is traditional to prepare a liqueur consumed after large meals as a digestive aid. In the past, a decoction with roots of mallow, fennel fruits, and dried figs was prepared in cases of fat cough. It was also used as a powerful amulet against the evil eye.

#### 3.2.2. *Hypericum perforatum* L.

*Hypericum* is native to the British archipelago, but today, it is widespread in the rest of the world and in all regions of Italy. It prefers dry soils, the edges of roads and fields, and sunny clearings, from the plain to the mountain [28]. It is widely spread in the Park of Cilento, Vallo di Diano and Alburni [24] and it is reported among the dyeing plants of Cilento, Vallo di Diano and Alburni Park [13].

From the ethnobotanical research, it emerged that the dialect name is *èriva rì San Giùuanni* (Table 2). It is mainly used for rituals related to the Catholic “day of ascension” and to prepare “St. John’s water”. On the eve of the night of 24 June (St John), it is necessary to collect a blend of herbs and flowers such as poppies, rosehip petals, mint, hypericum, lavender, chamomile, thyme, basil, sage, rosemary, mallow, walnut leaves, and wild fennel. At sunset, the herbs must be immersed in a basin of water and left overnight so that the dew settles on them. According to tradition, on the morning of the 24 June, it is necessary to wash the face and body with the scented water because it has the power to protect against disease, and to chase away the evil eye and bad luck. 

#### 3.2.3. *Lavandula angustifolia* Mill.

This lavender is native to the Western European area; in particular, the main areas of origin are ascribable to the Iberian and North African mountainous areas from where it spread throughout Europe. In Italy, it occurs spontaneously in the Ligurian–Piedmontese area and in the southern regions of the peninsula. The species is present at altitudes above 500–600 m a.s.l. and it is typical of the driest and sunny environments with calcareous soils [28]. In the Park of Cilento, Vallo di Diano and Alburni, there are three species of spontaneous lavender [24]: true lavender (*Lavandula angustifolia* Mill.), broad-leaved lavender (*Lavandula latifolia* Medik.), and wild lavender or steca (*Lavandula stoechas* L.). Among these, true lavender is by far the most represented and is widespread between Monte Cervati and Monte Bulgheria. 

The essential oil most frequently produced industrially is extracted from lavandin (L. × intermedia Emeric ex Loisel) [29]. 

Interest in lavender flowers and essential oil has been growing in recent years in different sectors: perfumery and cosmetics, food manufacturing, and aromatherapy [30,31,32,33,34]. Among other things, this plant plays a fundamental role in the survival of numerous species of wild bee in mountain areas [35].

The dialect name is *spicaddòsa* and it derives from the syncope of “odorous spica” to distinguish it from other non-scented ears such as those of the Poaceae [36]. Some others trace the dialectal name back to a popular linguistic deformation of “ear of bones” because in the past, the woody part of the plant was used for the treatment of bone fractures.

From the interviews, it emerged that lavender is one of the plants used to prepare St. John’s water and is still used today to decorate the saint’s statuettes. In Cilento, it is common to use bunches of lavender to perfume wardrobes and rooms and keep away moths that damage clothing. In the past, it was used to produce soaps with pork lard. 

The lavender that grows spontaneously on Monte Cervati was a well-known reality in the past and it represented an important economic factor thanks to the industrial process of the extraction of essential oils. Lavender harvested along the slopes of Monte Cervati was processed on site at Sanza in the years ranging from the First World War to the end of the 1960s. Hundreds of women in Sanza and neighboring countries were engaged in harvesting and distillation (Figure 4) to obtain the essential oil, which was sent to France for further final processing. Moreover, in those years, interventions were also made to improve the production of lavender. From the stories of older people, it emerged that the processing of lavender was an important source of income for many families. At the end of each harvesting campaign, the workers were paid for the work performed and a farewell party was organized for the occasion (“festa de ù Commito”) that included songs, dancing, and music. Oral stories of the women who had worked in the lavender harvest and photos of the period were collected and provided by the Association “Sanza Città della lavanda” [37]. It is still possible to see some remains of the lavender distillation plant at Sanza, in the Cornicello area (Figure 5).

Today there are numerous manifestations (conferences, events, and parties) that are organized to revive and enhance these traditions. In particular, the Association “Sanza città della lavanda”, active since 2002, proposes making it a reason for tourism development and enhancement of the territory, on an economic, cultural, and employment scale [37]. 

#### 3.2.4. *Malva sylvestris* L.

Mallow is found on uncultivated land, in trampled places, in ruderal environments, and at the edges of roads, and is also frequent in fields and meadows from 0–1600 m a.s.l. [28]. It is widely present in the Park of Cilento, Vallo di Diano and Alburni [24] and it is reported among the dyeing plants of Cilento, Vallo di Diano and Alburni Park [13].

Ethnobotanical use research revealed that its dialect name is *màleva* (Table 2) and in the popular tradition, decoctions are made for external use for the treatment of canker sores, eczema, and gingival and vaginal inflammation. A decoction of the roots is prepared for internal use for the treatment of bronchitis, cystitis, and inflammation of the respiratory tract. An infusion for gastritis and gastroesophageal reflux is prepared with the flowers. Grandparents used to rub the dry root on their teeth to fight tooth decay.

#### 3.2.5. *Matricaria chamomilla* L.

Common chamomile is native to the Eastern European area and was used in ancient times by both the Egyptians and the Greeks as a medicinal plant. Infrequent in stations located at altitudes above 300–400 m a.s.l., it is abundant in strong, dry, and skeleton-rich soils. It is found spontaneously throughout Italy, but it is not spread homogeneously in the different areas of the park [24]. In general, it adapts poorly to acid soils in which the plant produces a poor essence. It tolerates saline soils, and it grows discreetly in the presence of high pH and can be grown on highly calcareous soils. The whole plant gives off a pungent odor and has a bitter taste [28]. It is reported among the dyeing plants of Cilento, Vallo di Diano and Alburni Park [13].

From the interviews, it emerged that its dialect name is *cammumilla* (Table 2). In the past, dried flowers were used as pipe tobacco. Herbal teas were prepared as a calming drink to help with sleep and exhaustion, against eye inflammation, stomachache, and menstrual pain, and, together with fennel seeds, as a decoction against inflammation of the respiratory system. With the epigeal parts was prepared a poultice with mallow to mature boils and abscesses. In addition, the elders mixed food residues with soil and chamomile infusion to obtain a natural fertilizer.

#### 3.2.6. *Myrtus communis* L.

Myrtle is one of the main components of the Mediterranean scrub and is frequent on coasts, fixed dunes, and garrigue, where it lives in association with other characteristic elements such as mastic (*Pistacia lentiscus* L.) and rosemary (*Rosmarinus officinalis* L.). It forms dense wind-resistant bushes in mild-climate areas. It adapts very well to any type of soil, though it prefers a sandy substrate and tolerates drought well. Myrtle grows from sea level up to 500 m a.s.l. [28] and it is widespread throughout the park [24]. Both its leaves and berries are reported among the dyeing plants of Cilento, Vallo di Diano and Alburni Park [13].

From the interviews, it emerged that its dialect name is *murtédda* (Table 2). Traditionally, an alcohol extract is made with the berries for the treatment of colitis and a decoction is made for the treatment of sinusitis. With the leaves, on the other hand, a decoction for bleeding, dysmenorrhea, kidney disorders, and psoriasis and an infusion for the treatment of bronchitis and vaginal inflammation is made. 

In the past, the stems, deprived of the leaves, were used to make brooms, small objects, ladles, and handles. This plant was also used as fuel as it provides good firewood and excellent coal. The leaves, rich in tannins, were used for the tanning of hides or to aromatize olives in brine. Even today, liqueurs are prepared with the fruits. The use of sprigs of myrtle is still widespread to form garlands into which dried figs are inserted. Another use was that of “*muzzarella int’a’ murtedda*”, a cheese which was wrapped in myrtle leaves to preserve it, as well as taking on its aroma. Today, it is a Slow Food Presidium. 

#### 3.2.7. *Origanum heracleoticum* L.

Oregano is a perennial, bushy plant, which becomes semi-woody with aging, is extremely variable in appearance, and is pleasantly scented. Oregano grows in arid and sunny places, in sparse woods and stony places, and above all, in calcareous soils [28]. It is widespread in the Cilento Park [24].

From the interviews, it emerged that its dialect name is *arìgana* (Table 2). The drug consists of the aerial parts. In traditional medicine, it is used for internal use in cases of colds, flu, light fever, difficult digestion, aerophagia, stomachache, and painful menstruation. The flowering tops of the plant stimulate the secretion of gastric juices, aiding digestion, attenuating painful intestinal contractions, blocking intestinal fermentations, and eliminating gases [38]. 

The leaves and the flowering tops are dried and ground to prepare a spice that is particularly appreciated and used as a flavoring in various dishes of the Cilento tradition. To favor the release of the aromatic oils, the useful parts are macerated in oil to prepare condiments, especially with meat. In addition to these properties, oregano also acts as a calming agent, and is therefore useful in case of nervousness, insomnia, and tachycardia.

A curiosity is represented by the fact that the best harvesting areas of wild oregano are kept secret by its collectors and are transmitted from generation to generation. 

#### 3.2.8. *Rosmarinus officinalis* L.

Rosemary is an evergreen perennial woody shrub, and is very branchy with ascending, sometimes prostrate, but never really erect *habitus*. It is found in scrublands and garrigue from sea level up to 800 m a.s.l. It is a characteristic component of the Mediterranean scrub [28]. In the Cilento park, it is widespread both in the coastal areas, with a more prostrate shape, and in the internal ones, with a semi-prostrate shape [24].

Its dialect name is *rosamarìnu* (Table 2). Rosemary is used as a decoction, infusion, and medical wine (fresh buds). For internal use, it has digestive, antispasmodic, and carminative properties; stimulates diuresis and sweating; regulates the menstrual cycle; thins bronchial secretion; and calms convulsive coughs. In fact, rosemary contains active ingredients that give it anti-inflammatory, antiviral, antioxidant, antimicrobial, and anticancer properties [39].

The interviews carried out confirmed that rosemary is mainly used as a flavoring, but also to make decoctions against abdominal pain. Traditionally, branches blessed on palm day were placed at the entrance of the house, together with those of the olive tree, to protect against evil. In the past, it was used to inhale the smoke of burnt rosemary leaves, together with that of sage, during rites against the evil eye. It is a component of the “Water of Saint John”. In the past, the epigeal parts were put in boiling vinegar, left to cool, and used against gingivitis. Fumes were used to cure rheumatism.

### 3.3. Results of the Chemical–Physical Analysis of Aromatic Waters and Residual Distillation Waters

In Table 3 are reported the results related to the aromatic waters obtained for the six species that were distilled from fresh biomass (*Foeniculum vulgare*, *Lavandula angustifolia*, *Matricaria chamomilla*, *Myrtus communis*, *Origanum heracleoticum*, and *Rosmarinus officinalis*). All the aromatic waters kept the characteristic aroma of the species. For all aromatic waters, the acid pH values were confirmed to be in line with the literature data as the values fall within the typical range for hydrolates (between 5.36 and 6.74), making them compatible for common use as aromatic waters [40]. On the other hand, the EC values were between 30 (lavender) and 65 μS/cm (chamomile). As expected, the refractive index was equal to those reported in the literature for distilled water (1.33) [41].

Data related to the recovery of the residual waters from distillation process (Table 4) showed predominantly acidic pH, except for the fresh rosemary sample, which had a neutral pH of 7.02. With respect to aromatic waters, the residual waters showed higher values of EC between 948 μS/cm (chamomile) and 2340 μS/cm (oregano), which were still lower than the salinity limits of the solution circulating in the ground and of the water for irrigation.

The spectrophotometric analysis carried out on the aromatic waters highlighted variable spectra for each type of sample analyzed. All the spectra presented a peak at 290 nm with different values of absorbance; rosemary and chamomile also showed a peak at 310 nm (Figure 6).

On the contrary, the trends registered for the residual water from distillation showed limited differences up to 400 nm among the samples tested (Figure 7). All the residual waters showed a peak at 310 nm with an absorbance percentage of about 2.8; some other peaks were registered, the furthest of which were recorded for myrtle and oregano (Figure 7).

### 3.4. Results of Drying using the Traditional Method

Table 5 shows the results relating to the air-drying process carried out according to the traditional method of the area. Based on the data reported, the final moisture levels in the samples after the air-drying process were between 8.55 (hypericum) and 15.27% (lavender) (Table 5). The percentage of water after drying was slightly higher for the samples of lavender and mallow than for the others. In any case, the results can be considered satisfactory enough as they are close to standards required for good product conservation (10–12%) [8]. Lavender was the plant with the longest drying period, while mallow had the shortest period. For the other plants, an average duration of 20 days was recorded. 

From the quality control carried out on samples of dried biomass, it emerged that no foreign materials, molds, or damage from parasites were detected (Table 5). 

The analysis of the color through visual impact showed, for all samples, a bright color corresponding to that characteristic of the species and of the part of the drug. These results were confirmed via analysis carried out using the Munsell Atlas [21]. 

The olfactory analysis evidenced, for all samples of the aromatic plants, the odor characteristic of the species. 

### 3.5. Results of the Chemical–Physical Analysis of Aqueous Extracts and Anti-Germinative Activity

The results of the chemical–physical analysis of aqueous extracts are reported in Table 6, and from it, it emerges that all the extracts had an acid or sub-acid pH with slight variations between extracts of the same samples prepared using different techniques. Higher salinity was found for the extracts at 100 °C with respect to extracts at air temperature and those measured for residual distillation waters, as reported above. Additionally, in this case, the highest values were recorded for oregano with both methods of extraction.

The spectrometric analysis highlighted that the extracts showed a first peak at 310 nm with both methods of extraction and many other peaks at different distances, especially for extraction at 100 °C (Figure 8 and Figure 9). In particular, the extracts of myrtle and hypericum at 100 °C showed their last peaks at 600 nm (Figure 8).

Referring to the anti-germination activity, conducted on the different extracts of the dried samples with both extraction methods at air temperature and at 100 °C, very positive results were found (Table 7). Indeed, the number of germinated seeds for all the extracts were always lower than for the control (consisting only of water), except for the mallow extracted at air temperature. The lowest numbers of germinated seeds were found for oregano and lavender.

As for germinated seeds, the values of root elongation were always much lower than those of the control, thus demonstrating a greater effect of the extracts on root growth than germination. Consequently, all the samples, despite having demonstrated different results, proved effective in inhibiting radical elongation.

Both for the percentage of germination and for the radical elongation, the highest levels of inhibition were always recorded for the extracts at 100 °C compared to those obtained at air temperature.

### 3.6. Results of the Characterization of the Lavender Essential Oil

First of all, both oils tested had a typical color and a pleasant smell. Moreover, the GC and GC-MS analyses of the two samples resulted in the detection of 26 compounds, making up 99.8% of the total oil for Sanza lavender essential oil (SL) and 98.2% for commercial lavender oil (CL). All the compounds are listed in Table 8, according to their K_i_s (retention indices) on an HP-5MS column.

The two essential oils have been shown to contain a high amount of the two oxygenated monoterpenes linalool (45.0% in SL and 45.6% in CL) and linalyl acetate (19.9 and 14% for SL and CL, respectively), both considered among the most valuable components of lavender EO as they give a substantial contribution towards its aromatic characteristics [42]. Linalool was reported to be an important component of various aromatic plant essential oils. Its sedative, antioxidant, anti-inflammatory, and cardiovascular effects have been shown in various studies. Sabogal-Guáqueta et al. [32] suggested that chronic linalool administration induced a reduction in memory loss and emotional disturbances in people with Alzheimer’s. Caputo et al. [33] demonstrated the neuroprotective effect of linalool, but in general, of lavender essential oil, against neurotoxicity caused by the Aβ 1–42 oligomers, a key molecular factor in the neurodegeneration of Alzheimer’s disease. Furthermore, Caputo et al. [33], in a study conducted on rats, demonstrated a reduction in anxiety and stress levels after the administration of linalool.

Furthermore, the two essential oils SL and CL contain a similar amount of the oxygenated monoterpenes borneol (6.3 and 4.0%, respectively) and terpinen-4-ol (5.9 and 4.3%, respectively), but at the same time, while CL contains camphor in a high percentage (8.9%), this oxygenated monoterpene is present in a very small amount (0.9%) in SL. 

## 4. Discussion

This ethnobotanical research and these experimental results allow us to confirm that the valorization of spontaneous species, such as those covered in this study and/or others that are less known, represent an effective way to enhance and protect the territory.

Referring to the first aim of the present research—that is, to deepen our knowledge on the heritage of certain medicinal plants in an area of the Cilento, Vallo di Diano and Alburni Park—we can say that harvesting and using wild plants is a tradition strongly rooted in the rural communities of Cilento and is linked to the perception that the inhabitants of the community have of the environment in which they live. The uses detected in the present research mainly concerned traditional medicine and nutritional, ritual, or religious uses, and reflect the results obtained in analogous research in neighboring areas [43]. Moreover, the results were supported by the scientific world for the species examined, as briefly outlined below. Indeed, fennel has strong antioxidant activity thanks to the presence of large amounts of phenolic and flavonoid compounds [43] and is characterized by antibacterial, carminative, antispasmodic, and anti-inflammatory properties, making it useful in cases of digestive problems and airways diseases [44,45]. So, the seeds are used to prepare herbal teas with purifying and digestive action to treat stomach pain, colic, heartburn, and intestinal swelling. Given its antispasmodic and anti-inflammatory action, fennel tea also helps in cases of cough and airway diseases. Fennel seeds are also frequently used in cosmetics and cooking. 

The antiviral, antiretroviral, and antitumor activity of hypericum extracts has been demonstrated, suggesting the use of this plant in acquired immunodeficiency syndrome (AIDS) and cancer treatments [46]. Hypericum is also a valid remedy in cases of insomnia, anxiety, nervous breakdown, and mood swings related to seasonal changes or in the period of menopause. Hypericum oil has anti-inflammatory action, which makes it suitable for soothing massages in cases of muscle pain, back pain, rheumatic pain, neck pain, and sciatica [47]. Oleolite shows anti-inflammatory and soothing properties, which are useful in cases of skin burns, but also for red and inflamed skin, wounds, acne, and bedsores. Indeed, Lyles et al. [48] and Isacchi et al. [49] showed that hypericum oloite is useful in reducing the size of wounds and healing time thanks to its antibacterial properties. This oil is also capable of stimulating the formation and regeneration of epithelial tissue and is useful in the treatment of skin conditions such as psoriasis and atopic dermatitis. 

Lavender is probably one of the best known and most used essential oils and owes its fame to the relaxing, analgesic, and anti-inflammatory properties that characterize it. Malcolm et al. [50] demonstrated its anxiolytic action, and therefore, its usefulness in countering depression and sadness. The calming properties of lavender seem to be traced back to its ability to increase the activity of the parasympathetic system, which stimulates rest and relaxation. Furthermore, lavender is also neuroprotective; indeed, Koulivand et al. [51] showed that exposure to lavender aroma was beneficial in improving spatial memory in the case of Alzheimer’s. Lavender essential oil has proved useful in fighting pain, including chronic pain [52], in cases of trauma and bruises since it fights inflammation and edema formation [53]. Moreover, Malcolm et al. [50] demonstrated that lavender essential oil is antibacterial and antifungal thanks to its active ingredients such as linalool, geraniol, and eucalyptol, and helps in the wound healing processes. Lavender is an edible flower with high antioxidant content, including anthocyanins, carotenoids, and triterpenoids. Thanks to these substances, lavender helps to fight aging processes, but it is also proved useful in protecting the brain from degeneration and in counteracting high blood glucose values [54]. The scent of lavender, on the other hand, attracts bees who produce an excellent, aromatic, rare honey. 

The mallow drug consists of the flowers and leaves. In the medicinal field, mallow is an emollient; helps tissue regeneration; heals; calms irritation; and promotes the elimination of toxins. It is anti-inflammatory and acts as an excellent cleanser on the skin by removing impurities, counteracting acne, and even reducing dark spots on the skin [38]. In the cosmetic field, its leaves and flowers are used to prepare tonics, decongestant plasters, and beauty masks [55]. Additionally, food uses are reported by several authors [14,56,57]. 

The use of chamomile as a medicinal plant is handed down to us by Hippocrates, Galen, and Asclepius, and is a tradition that dates to the era of the ancient Greeks and Romans [58]. The flower heads of the common chamomile are used as a distillation material. Extracts prepared from *M. chamomilla* have been reported for their diverse range of pharmacological action, including cholesterol-lowering, antispasmodic, antiplatelet, anti-inflammatory, antimicrobial, antiviral, and antioxidant activities [59]. Furthermore, Della Loggia et al. [60] studied the sedative effects on the central nervous system of extracts prepared from this plant. The mother tincture of chamomile, thanks to the substances contained in it (terpenoids and flavonoids) [61], such as apigenin, rutin, and quercetin, shows anti-inflammatory, antispasmodic, sedative, and antibacterial properties [62,63,64,65]. In cosmetics, chamomile aromatic water is an excellent product suitable for delicate, sensitive skin that reddens easily, with irritation or inflammation, acne, eczema, or itching. This floral water, in fact, acts with anti-inflammatory, astringent, and antioxidant properties [66]. In addition, chamomile water is antimicrobial, soothing, and calms irritation and is an excellent skin tonic [66]. Chamomile flowers are also edible and can be added to salads, soups, or even fish dishes, such as seafood salads, to give a sweetish aroma and also add precious antioxidants. 

Hosseinzadeh et al. [67] evaluated the anti-inflammatory effect of aqueous and ethanolic extracts of myrtle, which showed activity against chronic inflammation and an analgesic effect through binding with opioid receptors or the release of endogenous opioid peptides. Ines et al. [68] evaluated its anti-proliferative and anti-genotoxic effects (cell protection from oxidative stress) through the activity of antioxidant enzymes and DNA repair enzymes. Mimica-Dukić et al. [69] studied its radical scavenger activity with an anti-mutagenic effect. Moradi et al. [70] evaluated its antiviral activity with an inhibitory effect on viruses before and after entering the host cell. Ferchichi et al. [71] confirmed its anti-ischemic hepatic effect through its antioxidant and radical scavenger activity. Shariati et al. [72] dealt with its ability to treat impotence through the inhibition of aromatase activity, the inhibition of 5-α reductase, and the inducer activity of cytochrome P450. Gholamhoseinian et al. [73] ascertained its antidiabetic effect by inhibiting the α-glucosidase enzyme. Tretiakova et al. [74] investigated its effect on cancer cell lines via the activation of caspase-3, -8, and -9, the splitting of poly PARP, the release of nucleosomes into the cytosol, and DNA fragmentation. Finally, Tayoub et al. [75] analyzed the insecticidal effect of the extract of this plant, which acts as a neurotoxin. In cosmetics, myrtle has proved very useful in cases of acne, but also irritated skin and even irritated gums and hemorrhoids, thanks to its anti-inflammatory, antiseptic, and antioxidant action [76]. The oil is used in perfumery, and in the preparation of soaps and cosmetics. A decoction of the leaves added to bath water has toning action. Due to the intense scent, myrtle flowers are used in the making of potpourri [38].

Oregano, has antioxidant, antiseptic, and antifungal properties [77,78], helping in cases of skin infections and mycosis such as athlete’s foot [79]. It brings mineral such as calcium, iron, magnesium, phosphorus, potassium, zinc, manganese, and selenium but also vitamins, such as vitamin C, vitamin A, and vitamins of the groups B, K, and E. Furthermore, it is rich in essential oils, with components such as carvacrol, thymol, and rosmarinic and caffeic acid [80]. In particular, carvacrol exhibits antiviral, antibiotic, and antimicrobial properties; thymol is anti-inflammatory, supports the immune system, and is fungicidal, proving effective in cases of candida of the oral cavity; and rosmarinic and caffeic acid are antioxidant compounds, useful against free radical damage [81,82,83]. Finally, oregano promotes integrity of the intestinal barrier; helps keep blood sugar under control; and protects heart health since it helps to prevent the oxidation of bad LDL cholesterol [79,84].

Studies about rosemary have shown that rosmarinic acid suppresses oral carcinoma by stimulating the activity of detoxifying enzymes, improves lipid and antioxidant peroxidation status, and decreases p53 and bcl-2 expression during DMBA-induced oral carcinogenesis [85]. Rosmarinic acid also plays an important role in the inhibition of various steps of angiogenesis and in the intracellular reduction of ROS [86]. In addition, rosemary protects the heart, improving its functionality; moreover, it has neuroprotective action thanks to carnosic acid, which seems to inhibit the formation of beta amyloid plaques, the accumulation of which is the main cause of Alzheimer’s [87]. de Oliveira et al. [39] showed that rosemary counteracts anxiety. Finally, a rosemary herbal tea can fight stress and fatigue; is an antioxidant; helps control blood sugar and weight; and protects the brain [88]. In cosmetics, studies have shown that rosemary helps to combat acne, is an antioxidant, is healing, anti-aging, and anti-inflammatory, and is believed to be able to fight the damage of the sun’s rays in skin cells [89,90,91]. The use of rosemary in cooking gives taste and healthy properties to dishes; the addition of rosemary to roast meat reduces the production of heterocyclic amines [92]; and rosemary has been shown to have a protective role against the toxicity caused by acrylamide, counteracting the oxidative stress induced by this substance on the nervous system [93].

The research results reported, in brief, justify the traditional uses and support the potential of the cultivation of the above plants both for medicinal uses and in the food sector for quality foods with health value.

In reference to the second purpose of the research, that is, to find other potential uses of studied species that are different from the traditional ones, the range of analyses conducted allows us to state that the collected species can be profitably employed in various uses. Indeed, data on the plant cover of the studied species highlight their contribution to soil and water conservation [24] and the ability of their root systems [26] to improve soil stability.

Regarding aromatic waters obtained from the harvested species, the chemical–physical analysis, in line with the literature data [18], confirmed that these are high-quality products that can be used very profitably in cosmetics, aromatherapy, and food. Additionally, scavenging activity determined against the 1,1-diphenyl-2-picrylhydrazyl radical (DPPH) was reported for basil aromatic water from Zaccardelli et al. [16]. Moreover, aromatic waters are considered easier to use than their respective essential oils, which can lead to migraines and skin and eye irritation. Indeed, they can be safely applied to the skin and can be taken orally [18]. Furthermore, aromatic waters can be used in cosmetics and in the food industry at a reduced cost compared to the corresponding essential oils [20], while retaining the aroma of the species. 

Regarding the potential use of water recovered from the distillation process, its use represent a contribution to increasing sustainability. These results highlight high anti-germinative activity and a low CE. For this reason, they can be used for weed control in cropping systems with low environmental impact. 

As for the spontaneous lavender that grows around Monte Cervati (Sanza, SA, Italy), the absence of camphor in the essential oil is considered a positive feature by both the cosmetic and pharmaceutical industries [94,95]. In fact, lavender EOs of higher quality, which are devoid of undesirable compounds such as camphor and used in the perfume industry, are produced by a few *L. angustifolia* cultivars. Moreover, the creation of an “industrial archeology park” in this area that goes beyond “remembrance” could become a very topical issue, and an opportunity for high valorization of the resources of the territory [37]. Regarding the drying process, quality control confirmed the validity of the traditional method (air drying) for ensuring good final moisture content, and the absence of foreign materials, molds, or damage from parasites. Similarly, in another of our studies [96], the results of the open-air drying of five medicinal species (*Althaea officinalis* L.; *Artemisia absinthium* L.; *Escholtzia californica* Chamisson; *Melissa officinalis* L.; and *Thymus vulgaris* L.) under a wooden structure for drying tobacco led to good-quality products with optimal humidity for conservation and minor impacts with respect to the artificial drying system, evaluated using the LCA procedure. Air drying can therefore represent a system that is easy to apply, even for small farm.

The present research, beyond deepening our knowledge on the heritage of medicinal plants in an area of the Cilento, Vallo di Diano and Alburni Park, also aimed to highlight that the collection and reproduction of spontaneous medicinal and aromatic species can have numerous implications of great interest. They contribute to the preservation of biodiversity by avoiding the excessive collection of spontaneous plants; make reproduction material available to nursery and cultivation companies and/or to anyone who has to intervene in protected areas; and enable the identification of local products to activate certified quality micro-chains. Furthermore, the knowledge and reproduction of local species contribute to identifying genotypes with greater adaptation to different pedoclimatic conditions and resistance to stress conditions, which is of great importance regarding current climate change, and in implementing the available medicinal heritage that can be used in various sectors in relation to the documented qualitative characteristics (food, herbal, the extraction of essential oils, environmental restoration, ornamental, excellent catering, etc.). Last but not least, the spread of aromatic plants contributes to the increase in pollinators and, in particular, of bees, also feeding the honey supply chain.

## 5. Conclusions

This research represents a contribution to the recovery of popular knowledge on some medicinal plants in an area of the Cilento, Vallo di Diano and Alburni National Park, and is an important opportunity to develop new activities that aim to enhance and protect the territory, with a view to rural development. The experimental results allow us to conclude that spontaneous medicinal plants could become potential sources of local economic development with uses in food and phytotherapeutics, but also in the agricultural sector, for example, for weed control.

## Figures and Tables

**Figure 1 plants-12-00465-f001:**
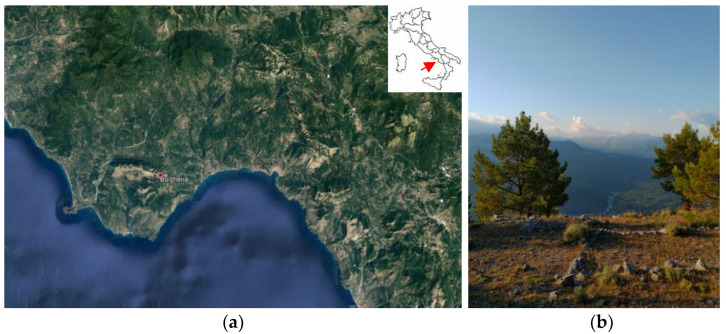
Cilento, Vallo di Diano and Alburni National Park, Monte Bulgheria. (**a**) Map from Google Earth; (**b**) Photo by G. Barile.

**Figure 2 plants-12-00465-f002:**
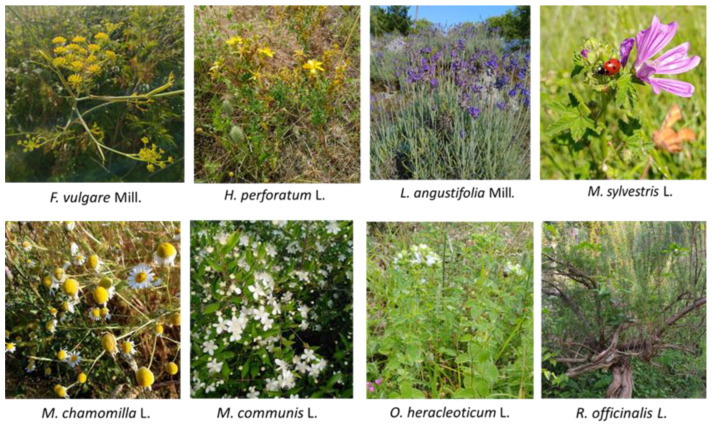
The spontaneous species collected. Photo by G. Barile (Monte Bulgheria).

**Figure 3 plants-12-00465-f003:**
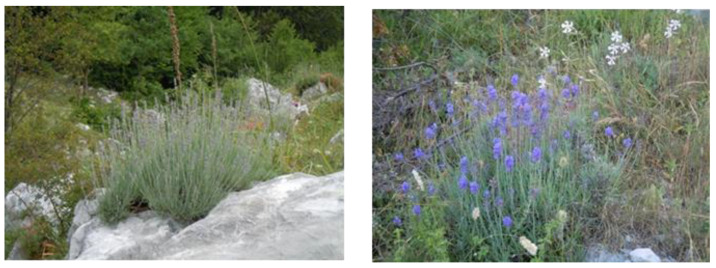
Lavender on Monte Cervati (Sanza, SA). Photo by E. De Falco.

**Figure 4 plants-12-00465-f004:**
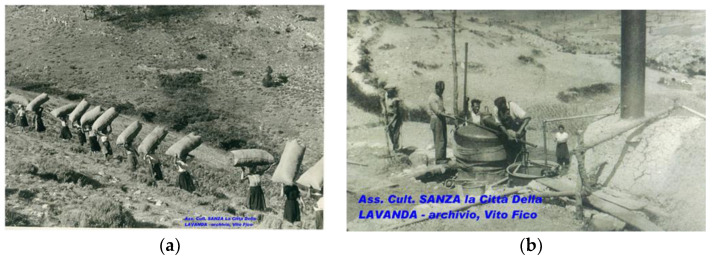
Historical documents: (**a**) women employed in the harvesting of lavender on Monte Cervati; (**b**) extraction of lavender essential oil in Sanza Municipality (SA). Photos made available by the Association “Sanza città della lavanda”.

**Figure 5 plants-12-00465-f005:**
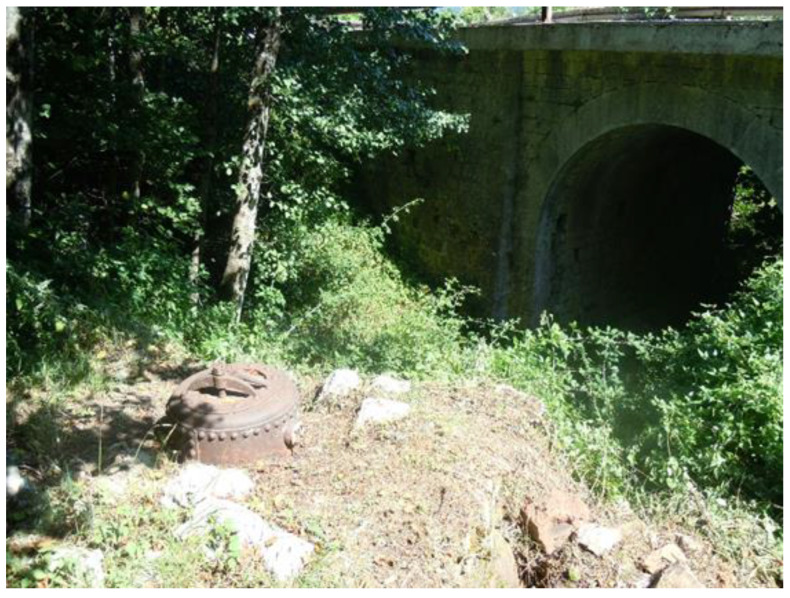
Some current remains of the distillation system at Sanza Municipality, Cornicello (SA). Photo by E. De Falco.

**Figure 6 plants-12-00465-f006:**
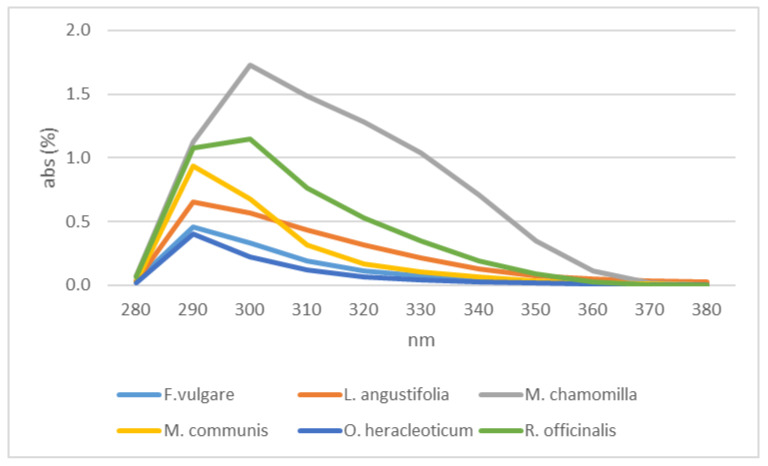
Aromatic waters absorbance.

**Figure 7 plants-12-00465-f007:**
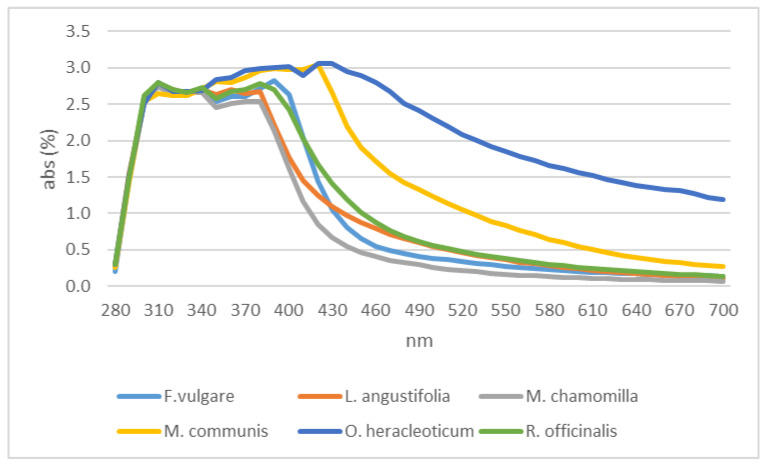
Residual waters absorbance.

**Figure 8 plants-12-00465-f008:**
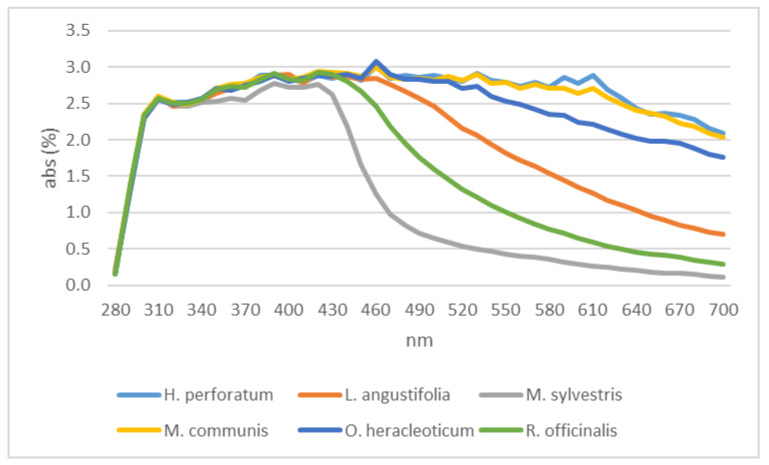
Absorbance of the extraction from dried product (100 °C).

**Figure 9 plants-12-00465-f009:**
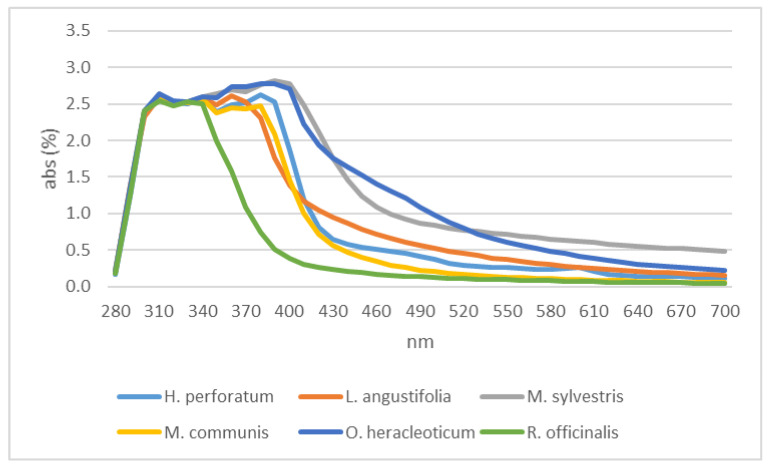
Absorbance of the extraction from dried product (air temperature).

**Table 1 plants-12-00465-t001:** Results relating to the relieves carried out during the harvesting of the plants.

Species	Geolocalization (Monte Bulgheria)	Harvesting Period	Phenological Stage	Vegetation Height (m)	Canopy cover Per Plant (m^2^)	Volume Estimate Per Plant (m^3^)	Leaf Weight (%)	Flower/Inflorescence Weight (%)	Moisture at Harvest (%)
*Foeniculum vulgare*	“Jacolina” 40°5′43.60″ N—15°24′48.96″ E	Third ten days of August	End of flowering—beginning of waxy ripening	1.67 ± 0.10	0.065 ± 0.037	0.106 ± 0.055	44 ± 8	56 ± 8	69.6 ± 0.2
*Hypericum perforatum*	“Altarino” 40°5′7.49″ N—15°24′49.41″ E	Third ten days of June	Flowering	0.45 ± 0.03	0.005 ± 0.003	0.002 ± 0.001	75 ± 4	25 ± 4	71.1 ± 0.2
*Lavandula angustifolia*	“Altipiano” 40°4′40.23″ N—15°23′57.88″ E	Third ten days of July	Flowering	0.67 ± 0.04	1.180 ± 0.313	0.786 ± 0.192	-	100 ± 1	62.9 ± 0.1
*Malva sylvestris*	“Jacolina” 40°5′43.60″ N—15°24′48.96″ E	Third ten days of April	Flowering	0.62 ± 0.12	0.138 ± 0.035	0.088 ± 0.039	69 ± 2	31 ± 2	80.4 ± 0.3
*Matricaria chamomilla*	“Jacolina” 40° 5′43.60″ N —15°24′48.96″ E	Third ten days of May	Flowering	0.28 ± 0.03	0.009 ± 0.005	0.003 ± 0.001	75 ± 4	25 ± 4	75.7 ± 0.2
*Myrtus communis*	“Morgialdo” 40°5′21.84″ N—15°25′2.18″ E	Third ten days of July	Flowering	1.72 ± 0.09	1.048 ± 0.279	1.820 ± 0.568	85 ± 5	15 ± 5	59.0 ± 0.1
*Origanum heracleoticum*	“Altarino” 40°5′7.49″ N—15°24′49.41″ E	Third ten days of June	Beginning of flowering	0.38 ± 0.07	0.088 ± 0.058	0.037 ± 0.029	80 ± 3	20±3	70.0 ± 0.2
*Rosmarinus officinalis*	“Altarino” 40°5′7.49″ N—15°24′49.41″ E	Third ten days of June	Vegetative stage	1.36 ± 0.14	2.776 ± 0.592	3.830 ± 1.187	100 ± 1	-	63.4 ± 0.2

**Table 2 plants-12-00465-t002:** Results relating to the local and common names per the species under study.

Species	English Common Name	Local Name
*Foeniculum vulgare*	Wild fennel	Finùcchiu sirvàticu
*Hypericum perforatum*	Hypericum	Èriva rì San Giùuanni
*Lavandula angustifolia*	True lavender	Spicaddòsa
*Malva sylvestris*	Mallow	Màleva
*Matricaria chamomilla*	Common chamomile	Cammumilla
*Myrtus communis*	Myrtle	Murtédda
*Origanum heracleoticum*	Oregano	Arìgana
*Rosmarinus officinalis*	Rosemary	Rosamarìnu

**Table 3 plants-12-00465-t003:** Aromatic waters: results of the chemical–physical analysis and refractive indices.

	pH	EC (μS/cm)	Refractive Indices (n)
*Foeniculum vulgare*	6.34	44	1.3333
*Lavandula angustifolia*	5.86	30	1.3335
*Matricaria chamomilla*	6.74	65	1.3332
*Myrtus communis*	6.09	50	1.3333
*Origanum heracleoticum*	6.25	46	1.3332
*Rosmarinus officinalis*	5.87	35	1.3333

**Table 4 plants-12-00465-t004:** Results of the chemical–physical analysis of the residual distillation waters.

	pH	EC (μS/cm)
*Foeniculum vulgare*	6.44	1285
*Lavandula angustifolia*	5.46	1949
*Matricaria chamomilla*	6.23	948
*Myrtus communis*	4.63	1156
*Origanum heracleoticum*	5.43	2340
*Rosmarinus officinalis*	6.13	1552

**Table 5 plants-12-00465-t005:** Results related to the air drying of the samples according to traditional methods.

Species Harvested	*Hypericum perforatum*	*Lavandula angustifolia*	*Malva sylvestris*	*Myrtus communis*	*Origanum heracleoticum*	*Rosmarinus officinalis*
Drying duration (days)	20	27	16	21	20	20
Moisture (%)	8.55	15.27	12.31	10.66	11.66	9.09
Foreign materials	NO	NO	NO	NO	NO	NO
Foreign plant materials (traces)		Rosemary		Lavender		Oregano
Molds	NO	NO	NO	NO	NO	NO
Pest damage	NO	NO	NO	NO	NO	NO
Munsell color						
of the leaf blade	10Y 4/4		7.5Y 5/6	10Y 5/4	2.5GY 5/2: 2.5GY 5/4; 2.5GY 6/4; 2.5GY 6/6	10Y 6/2; 10Y 6/4
of the flowers	2.5Y 8/10	10PB 4/4; 10PB 3/8	2.5P 3/6	7.5Y 9/4		
Visual impact color						
of the leaf blade	Dark green		Bright light green	Dark green	Bright light green	Dark green with light shades
of the flowers	Dark ocher yellow	Blue/deep purple	Bright purple	Light yellow		

**Table 6 plants-12-00465-t006:** Results of the chemical–physical analysis of the different extracts.

Samples	Extracts at 100 °C		Extracts at Air Temperature	
	pH	CE (μS/cm)	pH	CE (μS/cm)
*Hypericum perforatum*	5.38	3600	5.30	1000
*Lavandula angustifolia*	5.68	3640	5.81	1182
*Malva sylvestris*	6.10	3940	6.80	2000
*Myrtus communis*	4.80	2700	5.60	741
*Origanum heracleoticum*	5.72	4170	5.67	1513
*Rosmarinus officinalis*	6.26	1506	6.20	862

**Table 7 plants-12-00465-t007:** Comparison of the anti-germination activity of the various extracts of the different analyzed species. Different letters within the same column indicate significant differences among aqueous extracts, according to ANOVA combined with Tukey post-hoc test at *p* = 0.05.

	Extract at Air Temperature		Extract at 100 °C	
Sample	Germinated Seeds (%)	Root Length (mm)	Germinated Seeds (%)	Root Length (mm)
Water (control)	93.3 ± 0.6 a	22.9 ± 0.3 a		
*Malva sylvestris*	90.0 ± 1.0 a	1.5 ± 0.4 c	66.6 ± 3.5 c	
*Hypericum perforatum*	83.3 ± 2.1 b	10.5 ± 0.3 b	83.3 ± 1.2 b	
*Myrtus communis*	83.3 ± 1.5 b	6.9 ± 1.0 b	83.3 ± 0.6 b	2.04 ± 0.9 c
*Lavandula angustifolia*	53.3 ± 1.5 c	2.5 ± 0.1 c	36.6 ± 2.1 d	
*Origanum heracleoticum*	30.0 ± 1.0 d	1.8 ± 0.7 c	60.0 ± 2.0 c	0.96 ± 0.3 c
*Rosmarinus officinalis*	76.6 ± 1.5 b	14.5 ± 3.5 b	60.0 ± 1.0 c	2.49 ± 0.3 c

**Table 8 plants-12-00465-t008:** Comparison of the composition of Sanza lavender essential oil (SL) and of a commercial one (CL).

K_i_ ^a^	K_i_ ^b^	Compound	Sanza Lavender (SL) (%) ^c^	Commercial Lavender (CL) (%) ^c^
993	1173	Myrcene	0.5	1.3
1030	1203	Limonene	T	0.6
1034	1213	1,8-Cineole	-	4.1
1040	1232	(*Z*)-β-Ocimene	0.2	0.7
1049	1265	(*E*)-β-Ocimene	0.3	0.8
1076	1477	*cis*-Linalool oxide (furanoid)	0.5	0.2
1080	1436	*trans*-Linalool oxide (furanoid)	0.4	T
1098	1553	Linalool	45.0	45.6
1109		Octenol acetate	1.1	-
1145	1532	Camphor	0.9	8.9
1167	1718	Borneol	6.3	4.0
1176	1611	Terpinen-4-ol	5.9	4.3
1189	1706	α-Terpineol	6.6	5.9
1218	1806	Nerol	0.8	-
1237	1847	Geraniol	0.1	-
1259	1665	Linalyl acetate	19.9	14.0
1284	1597	Bornyl acetate	0.2	-
1287	1599	Lavandulyl acetate	0.8	3.2
1358	1601	Neryl acetate	2.1	1.3
1388	1731	Geranyl acetate	3.9	2.2
1418	1612	(E)-β-Caryophyllene	1.0	0.4
1440	1688	(E)-β-Farnesene	T	-
1477	1726	Germacrene D	0.2	0.6
1579	2208	Caryophyllene oxide	0.4	-
1652	2255	α-Cadinol	2.7	-
1662	2153	α-Bisabolol	-	1.0
		**Total**	**99.8**	**98.2**

^a^ K_i_: retention index on an HP-5MS column; ^b^ K_i_: retention index on an HP-INNOwax column; ^c^ t: trace, less than 0.05 %.

## Data Availability

Not applicable.

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
