# Peer review of "Spontaneous Officinal Plants in the Cilento, Vallo di Diano and Alburni National Park: Tradition, Protection, Enhancement, and Recovery"

_plants, 2023, doi:10.3390/plants12030465_

Round 1
Reviewer 1 Report
I do not understand the purpose of this paper. The aim states: "...the research aimed to deepen the knowledge on the heritage of medicinal plants of an area of the Cilento, Vallo di Diano and Alburni Park to detect the potential production and the traditional uses to enhance and protect the territory also through the development of potential forms of local economies also through the recovery of the industrial archeology."
To start with, that is very poor English. Split this one sentence into 3 or 4 short sentences that explain the aims of the research.
The paper focuses on 8 medicinal plant species. No indication why those 8 were selected. Random? Are those the only species still traditionally collected / used in the area? Explain why you selected those 8.
Then ethnobotanical knowledge is gathered from local elders. How many people were interviewed? 3? 20? No indication. What questions were asked? That is important information. You state that culturally polluted information was excluded, you conclude that the knowledge is transmitted from generation to generation and 5 pages of results are dedicated to the traditional uses. Then I would like to see evidence of the kinds of questions you have asked, what information was collected in these interviews, how many people were interviewed.
Next a range of analyses are carried out on aromatic waters and water extractions from dried herbs. This gives information on pH, electrical conductivity, spectrophotometric spectra, etc. Why is this important? What information does this provide? How are these results related to the traditional uses, or how are they important for traditional uses? That is not explained anywhere in the results or the discussion. Does this provide any information on the potential medicinal properties of the plants, or on other potential uses? There's only a vague statement in the conclusion that the results indicate these are 'high quality products' for use in cosmetics.
Then there's an experiment on the water extracts reducing the germination of cress seeds. What is the purpose of this? What does it tell us? Is there a relation with the traditional uses? In the conclusion there's mention of potential use for weed control. Is that the purpose of the test? Then tell us more about what the results tell us.
The discussion focuses entirely on a variety of uses of the 8 species as reported in literature. There is no link at all with the results of your own experiments and analysis. So what was the purpose of those tests/experiments?
So the paper really consists of an in-depth literature study of the traditional and contemporary uses of 8 plant species; and a range of tests / analysis carried out on those same 8 species, but without there being any connection between those 2 parts. If there is a connection, make it clear.
Reviewer 2 Report
The article “Spontaneous officinal plants in the Cilento, Vallo di Diano and Alburni National Park: between tradition, protection, enhancement, and recovery” describes ethnobotanical knowledge on the traditional uses of medicinal plants of the Cilento, Vallo di Diano and Alburni National Park (Salerno province). Furthermore, the authors tried to evaluate the productive potential of medicinal plants to increase possible uses for recovering and enhancing the studied territory.
The main strength of this paper is that it addresses an insufficiently explored question and finds a novel solution based on a carefully selected set of ethnobotanical data for analysis. As such, this manuscript represents an ethnobotanical study which will almost certainly influence future regional medicinal plants research as a starting point. However, this study requires additional improvement.
Major points to consider in subsequent versions:
-
The map of the study area would be most welcome.
-
Throughout the manuscript, the authors used photographs as illustrations (Figures 1, 2, 4, 5, 6). The authors should include more precise information about them. For example, the copyrights, date and place where the photos were taken are listed not everywhere.
-
Chapter 2.2 Ethnobotanical uses need additional improvement: it is unclear whether the authors conducted ethnobotanical fieldwork. For example, how many people were interviewed? Where were the interviews conducted? I would suggest describing the data collection process more precisely.
-
Recording of traditional local ecological knowledge is very significant nowadays. In addition to documenting ethnobotanical uses of wild plants, this also includes recording local plant names. Therefore, I would ask about adding in Table 1 an additional column with the Local name (s) of the species and the number of herbarium specimens.
-
In the title of chapter 3.2, the authors are divided into ethnobotany and tradition. I would suggest splitting them together because ethnobotany means traditional uses of plants.
-
Including information about ethnobotanical uses of medicinal plants is highly welcome for the purposes stated by the authors (Chapters 3.2.1 - 3.2.8). But it is not clear why the authors have chosen these species for analysis. Is it only the taxons that were recorded in interviews? Perhaps, including an additional table, “Ethnobotanical uses” of studied plants, will improve this issue.
Finally, the paper “Spontaneous officinal plants in the Cilento, Vallo di Diano and Alburni National Park: between tradition, protection, enhancement, and recovery” might be accepted after major revision.
Round 2
Reviewer 1 Report
Thanks for the revisions made.
Author Response
Dear Reviewer 1,
thanks again for the valuable suggestions and advice you have given us, which have greatly improved the manuscript.
Best regards
Reviewer 2 Report
The authors of the manuscript “Spontaneous officinal plants in the Cilento, Vallo di Diano and Alburni National Park: between tradition, protection, enhancement, and recovery” have clarified several questions I raised in my previous review. However, two areas still need revisions:
-
Line 262. I would suggest changing “Results of the ethnobotany research” to “Results of ethnobotanical research”
-
Line 266. Table 2. I would suggest changing “Dialectal name” to “Local name” and “Common name” to “Common name in Italian”.
Finally, the manuscript “Spontaneous officinal plants in the Cilento, Vallo di Diano and Alburni National Park: between tradition, protection, enhancement, and recovery” might be accepted after minor revision.
